# Protection and Restoration of Damaged Hair via a Polyphenol Complex by Promoting Mechanical Strength, Antistatic, and Ultraviolet Protection Properties

**DOI:** 10.3390/biomimetics8030296

**Published:** 2023-07-09

**Authors:** Hyun Jeong Won, Tae Min Kim, In-sook An, Heung Jin Bae, Sung Young Park

**Affiliations:** 1Department of Chemical and Biological Engineering, Korea National University of Transportation, Chungju 27469, Republic of Korea; dnjsswh@ut.ac.kr (H.J.W.); ktm1120@a.ut.ac.kr (T.M.K.); 2Korea Institute of Dermatological Sciences, Seoul 05836, Republic of Korea; anis@skinresearch.co.kr; 3MODAMODA Corporation, Ltd., Seoul 05546, Republic of Korea; 4Department of IT and Energy Convergence (BK21 FOUR), Korea National University of Transportation, Chungju 27469, Republic of Korea

**Keywords:** polyphenol, UV protection, antistatic, tannic acid, gallic acid, caffeic acid

## Abstract

In this study, we developed a hair-coating polyphenol complex (PPC) that showed ultraviolet (UV) protection properties, antistatic features, and the capability to enhance the mechanical strength of damaged hair. PPCs prepared with different ratios of tannic acid (TA), gallic acid (GA), and caffeic acid (CA) simultaneously increased the self-recovery of damaged hair by protecting the cuticle. PPC prevented light from passing through the damaged hair during exposure to UV radiation. Moreover, surfaces coated with PPC1 (TA:GA:CA, 100:20:0.5) exhibited a higher conductivity than surfaces coated with PPCs with other ratios of TA, GA, and CA, with a resistance of 0.72 MΩ. This influenced the antistatic performance of the surface, which exhibited no electrical attraction after being subjected to an electrostatic force. Additionally, damaged hair exhibited a significant increase in durability and elasticity after coating with a PPC1-containing shampoo, with a tensile strain of up to 2.06× post-treatment, indicating the recovery of the damaged cuticle by the PPC complex. Furthermore, PPC1-containing shampoo prevented damage by scavenging excess reactive oxygen species in the hair. The combination effect promoted by the natural PPC offers new insights into hair treatment and paves the way for further exploration of hair restoration technology.

## 1. Introduction

Hair dyeing and bleaching have become a recent trend that harms the hair, especially the hair shaft, decreases its mechanical strength, and makes it prone more to breakage [1]. Moreover, regular dying and/or bleaching of hair causes unfavorable growth patterns, reduced glossiness, and difficulty in management; thus, affecting the visual perception of the hair, which could further impact a person’s confidence level [2,3]. Additionally, habitual usage of a hair drier can cause roughness, dryness, and color loss due to a rapid decrease in hair moisture after exposure to warm air [4]. Natural phenomena often lead to changes in hair structure and color. For example, ultraviolet (UV) radiation affects keratin, lipids, and melanin concentration in hair [5,6]. Subsequently, post-UV exposure, accumulation of reactive oxygen species (ROS) is observed, which is further exacerbated due to prolonged exposure. Additionally, UV radiation induces photodegradation of hair proteins that increases cuticle size and induces cavity formation in the hair shaft [7,8]. Hence, a damaged hair shaft increases fragility and vulnerability and decreases the structural integrity of the hair [9]. Moreover, since external stimuli mostly damage the hair cortex and cuticle, preserving these regions may be beneficial in preventing future environmental or chemical harm.

Spider silk is a natural high-performance fiber with great mechanical strength compared to synthetic materials [10]. This high mechanical performance is due to the presence of intermolecular interactions between β-sheets and -turns of the protein strands of spider silk, such as hydrogen bonding, hydrophobic interactions, van der Waals forces, and self-assembly stacking [11]. Polyphenol compounds have attracted the interest of researchers over the last decade owing to their multifunctional properties, such as strong adhesiveness, excellent biocompatibility, and remarkable coating ability [12,13,14]. Polyphenols contain a phenyl chemical structure linked to two or more phenolic hydroxyls, which is found in many organisms [15,16,17]. Owing to the structure of PPCs that contain a lot of *sp*^2^ orbitals, spider silk-inspired materials based on π-π stacking and hydrogen bonding can be obtained. Furthermore, polyphenols demonstrate notable antioxidant, immunological modulation, and anti-radiation properties, making them viable candidates for use in pharmaceutical, food, and cosmetic products [18,19,20]. Tannic acid (TA), a polyphenol isolated from plants, provides UV protection owing to its ability to absorb UV radiation [21]. Additionally, tannic acid can enhance the electrical conductivity of a material owing to its abundance of surface groups and ability to stabilize molecular networks [22,23]. Other phenolic compounds that have shown promising performance are gallic (GA) and caffeic (CA) acids. In particular, GA imparts plasticizing properties that improve durability, reduce brittleness and flexibility, and enhance elasticity [24]. In contrast, CA demonstrates good antioxidant activity by preventing lipid and protein oxidation and an increase in scavenging ROS [25,26]. Therefore, a multipurpose integrated system can be developed by utilizing the full potential of these polyphenols.

Therefore, in this study, we prepared a hair-coating polyphenol complex (PPC; TA/GA/CA) by combining tannic, gallic, and caffeic acids to provide protection from UV radiation, minimize electrostatic effects, and improve the mechanical strength of hair by covering the hair cuticle. Based on the distinct advantages of each phenolic compound, an integrated multipurpose material capable of protecting and enhancing hair self-recovery was developed. The PPC was introduced into a common shampoo, and the recovery of damaged hair was assessed. The PPC coating on the hair cuticle was formed via physisorption. Additionally, hair quality was enhanced by reducing the pore size in the hair shaft and providing additional strength and rigidity. Moreover, the presence of PPC reduced ROS levels in the scalp and hair, which could improve the hair’s mechanical strength and morphology.

## 2. Materials and Methods

### 2.1. Material and Characterization

The MODAMODA Corporation (Seoul, Republic of Korea) supplied the Pro-Change Black Shampoo^®^ (containing water, sodium C14-16 olefin sulfonate, disodium laureth sulfosuccinate, glycerin, sodium cocoyl isethionate, lauryl hydroxysultaine, butylene glycol, lauryl glucoside, 1-2-hexanediol, *Sesamum indicum* (sesame) seed extract, *Morus nigra* fruit extract, *Tuber melanosporum* extract, *Nigella sativa* seed extract, *Prunus serotina* (wild cherry) fruit extract, *Morus alba* bark extract, niacinamide, *Camelia sinensis* leaf extract, 1,2,4-trihydroxybenzene, salicylic acid, citric acid, *Mentha piperita* (peppermint) oil, sodium chloride, PPG-3 caprylyl ether, guar hydroxypropyltrimonium chloride, panthenol, polyquartenium-10, ethylhexylglycerin, caprylyl glycol, menthol, caramel). TA, GA monohydrate, CA, phosphate-buffered saline (PBS), 2,2-diphenyl-1-picrylhydrazyl (DPPH), hydrogen peroxide (H_2_O_2_), and sodium hydroxide (NaOH) were purchased from Sigma-Aldrich (St. Louis, MO, USA). Human hair samples (100% human hair, beige color; Milbon level 19, Wella chart 15) were purchased from Yeosin (Seoul, Republic of Korea).

UV–visible (vis) spectra were obtained using an Optizen 2020 UV spectrometer (Mecasys, Daejeon, Republic of Korea). Particle size was measured using a Zetasizer Nano (Malvern Panalytical, Kassel, Germany). Photoluminescence spectra were obtained using a FluoroMate fluorometer (Scinco, Seoul, Republic of Korea). Scanning electron microscopy (SEM) images were recorded using a JSM-6700F microscope (JEOL, Tokyo, Japan). Thermal gravimetric analysis (TGA) was performed using an SDT 650 thermal analyzer (TA Instruments, New Castle, DE, USA) at a heating rate of 10 °C/min in an N_2_ environment. Color was observed using a colorimeter (Konica Minolta, Tokyo, Japan). Fourier transform infrared (FT-IR) was investigated using Nicolet iS10 series (Thermo-Fisher, Waltham, MA, USA). The static contact angle was measured using a KRUSS DSA100 instrument. Electrical resistivity was quantified using a Keithley 2450 source meter (Tektronix, Inc., Beaverton, OR, USA). Confocal laser-scanning microscopy (CLSM) images were recorded using an ECLIPSE Ti2-E confocal microscope (Nikon, Tokyo, Japan). Tensile mechanics were analyzed using a texture analyzer (SurTA 1A; Chemilab Co., Seoul, Republic of Korea).

### 2.2. Preparation of PPC Nanoparticles and PPC-Incorporated Shampoo

To synthesize PPC, TA, GA, and CA were dissolved in 1 mL of PBS (pH 8.5) at various weight ratios (Table 1) and stirred for 12 h at room temperature. Thereafter, the PPC was added to 30 g of shampoo, mixed for 30 min at 40 °C, and labeled as PPC shampoo.

### 2.3. Investigation of the Electrical Performance and Coatability of the PPC Nanoparticles

To investigate electrical performance and coating ability, a PPC-coated substrate was prepared on a silicon (Si) wafer with dimensions of 1 cm × 1 cm (length × width). The PPC was dissolved in double distilled water (DDW) to a concentration of 10 mg/mL. The Si wafer was immersed in the PPC solution for 12 h and dried before measurement. The electrical resistance of the coated surfaces was measured using a Keithley 2450 source meter at a bias voltage of 1 V using the two-electrode technique. The static contact angle was measured using a KRUSS DSA100 instrument and analyzed using the Cecil drops method. Each measurement was performed thrice (*n* = 3).

### 2.4. Evaluation of Anti-Electrostatic and Mechanical Properties of PPC-Coated Hair Samples

The PPC shampoo performance and the method for applying the shampoo were investigated using hair samples. The PPC shampoo was applied to the hair on all sides at least 30 times and allowed to coat the hair for 30 min. The hair samples were then washed with water, incubated at room temperature for 1 h, and dried using a hair dryer. For the hydrogen peroxide (H_2_O_2_) condition, the hair samples were immersed in H_2_O_2_ solutions of different concentrations (0, 6, and 9%) for 12 h and then washed with running water for 1 min before PPC shampoo treatment. For the heat treatment, the hair samples were dried 30 times (one-directional swipe) with a hair straightener at a temperature of 220 °C (44 W) before PPC shampoo treatment. UV protection analysis was performed by irradiating the hair sample at 302 nm for 72 h. Initially, the hair samples were dyed using a red dye (Allura Red AC 40) as a marker of color degradation. The dye was initially applied to the hair 30 times (all sides of the hair) and allowed to coat the hair for 30 min. The hair was then washed with running water and left to naturally dry for 1 h. Thereafter, the PPC shampoo was applied to the hair, and the hair was irradiated with UV light and investigated using a colorimeter (Konica Minolta, Tokyo, Japan).

To evaluate anti-electrostatic performance, a rubber balloon was filled with air, rubbed onto a cotton fabric thrice, and placed near the hair before capturing images. The tensile mechanics of the coated hair samples were measured at a distance of 13.5 cm between the static (bottom) and dynamic (top) grips at a displacement rate of 10 mm/s.

### 2.5. Application of the PPC for ROS Scavenging

The ROS scavenging activities of the PPC nanoparticles and PPC shampoo samples were investigated. A DPPH solution (0.2 mM, 1.5 mL) was added to the PPC nanoparticles (1.5 mL) and, after placing the mixture in a 37 °C chamber for 30 min, the absorbance was measured at 517 nm to calculate the inhibition percentage. For another measurement, 3 mL of 0.1 mM DPPH was added to 3 mL of PPC shampoo, and the samples were incubated at 37 °C for 30 min before measuring the solution using UV–vis spectroscopy. DPPH inhibition was calculated using the following equation:$$DPPH\;inhibition\;\left(\%\right)=\frac{\left(A_{std}-A_{sam}\right)}{A_{std}}\times 100\%,$$
where *A_std_* is the absorbance of the DPPH standard and *A_sam_* is the absorbance of the sample.

## 3. Results and Discussion

### 3.1. Design of PPC Shampoo for Hair Protection and Repair

PPC shampoo was fabricated by incorporating TA/GA/CA into a common shampoo formula. The combination of TA/GA/CA played a pivotal role in hair protection (e.g., from UV radiation, hair dyeing, and hot hair drying processes) and repair by enhancing the mechanical and electroconductivity, durability, and UV protection properties of the hair owing to their individual characteristic properties [21,22,23,24,25,26]. The PPC formed a graphene-like structure via hydrogen bonding and π-π stacking between each phenolic species after complexation using PBS (pH 8.5), resulting in improved mechanical strength, which plays a vital role in hair restoration (Figure 1a) [15]. This graphene-like structure also increased the electroconductivity of the hair shaft that efficiently dissipated the electrostatic charges of sticky hair, allowing an antistatic effect on PPC-treated hair [27]. Moreover, the antioxidant properties of the PPC reduced the levels of ROS in hair follicles generated by continuous UV exposure, hair bleaching, or dyeing processes, and reduced trace metal levels in the scalp. Additionally, owing to its rich polyphenol structure, the PPC could easily coat the hair cuticle; they were stable on the hair cuticle for an extended period. Thus, providing prolonged hair protection and repair by protecting from the photobleaching effect of UVB radiation, even after frequent continuous washing [28]. Furthermore, PPC combined with the Pro-Change Black shampoo will provide strengthened UV protection and mechanical girth to the hair that could not be achieved by other conventional hair dyes or graphene-based shampoos (Figure 1b).

### 3.2. Characterization of the As-Synthesized PPCs

The TA/GA/CA ratios were varied to create PPC1, PPC2, PPC3, and PPC4, as shown in Table 1, and each PPC particle size was determined using a dynamic light scattering spectrometer. The average diameters were found to be 850 ± 10 nm for PPC1, 650 ± 20 nm for PPC2, 400 ± 12 nm for PPC3, and 350 ± 16 nm for PPC4. The optical and structural properties of these synthesized PPCs were evaluated using a UV–vis spectrometer.

The absorption peak at 275 nm for PPC1, PPC2, and PPC3 and a peak at 245 nm for PPC4 indicated the presence of π-π* attributed to the aromatic group of the polyphenols (Figure 1a) [29,30,31]. Moreover, the broad absorption spectrum of PPC1, which ranged from 380 nm to 600 nm, further indicated the capability of PPC1 to absorb a wide range of light spectra owing to the graphene-like π-π stacking structure. Thus, PPC1 will be suitable for hair protection from sunlight, which is known to contain a broad light spectrum (Figure 1a, inset). The shifting of the π-π* absorption peak also indicated the effect of intermolecular interactions, such as hydrogen bonding and π-π stacking. The PPC samples were further subjected to UV protection tests by assessing UV light transmission on a paper screen after passing the light through the PPC solutions and DDW as a control. Figure 1b showed that UV light was transmitted to the paper screen after passing through a glass cuvette containing DDW. All PPC samples (PPC1–4) demonstrated UV protection properties, confirmed by the absence of UV light transmission on the paper screen behind the PPC solutions. Additionally, the photoluminescence spectra showed that PPC1 emitted blue fluorescence due to the presence of the π-π stacking structure, which served as a fluorophore (Appendix A).

UV protection was achieved by the PPC samples owing to the potential of polyphenols (particularly TA and CA) to absorb a broad spectrum of UV light. Therefore, PPC could be expected to protect hair from direct exposure to UV radiation received from sunlight when applied via shampoo [2]. The structural characteristics of the PPCs were assessed using Fourier transform infrared spectroscopy (Figure 2). PPC1, PPC2, PPC3, and PPC4 possessed a broad oxygen–hydrogen (O-H) stretching peak centered around 3300–3600 cm^−1^ due to the polyphenols. A spectral shift of the O-H stretching peak occurred in PPC1 compared with PPC2, PPC3, and PPC4. This finding corresponded with the availability of more hydrogen bonds in PPC1 than in PPC2–4, due to the ratio of TA, GA, and CA in PPC1. This was further supported by TGA data, which showed an increase in the degradation temperature of CA (217–240 °C) [32], GA (325–400 °C) [33], and TA (380–600 °C) alone [34], compared to when they were part of the synthesized PPC1 complex (203–289 °C, 292–493 °C, and 400–646 °C for CA, GA, and TA, respectively). This was related to the hydrogen bonding in PPC1 (Appendix A).

PPCs were expected to coat the hair and remain stable for a long time, owing to their adhesive properties even after washing several times during the application of the PPC shampoo [29,35,36]. Hence, it was necessary to determine the stability of the PPCs after surface coating, which was evaluated by coating each PPC sample on the surface of an Si wafer and comparing the conductivity and contact angle of each PPC-coated surface without (0 washing) and with 10 washes using DDW. Based on source meter measurements (Figure 3a), the unwashed PPC1-coated surface showed lower resistance (0.72 MΩ) than the unwashed surfaces coated with PPC2 (1.33 MΩ), PPC3 (1.11 MΩ), or PPC4 (2.41 MΩ) and the bare Si wafer (0.874 MΩ). This lowered resistance was achieved owing to a more conductive network produced by π-π stacking between TA, GA, and CA in PPC1. Lower resistance or higher conductivity is crucial for obtaining an antistatic effect through efficient electrostatic charge dissipation [27]. After each PPC-coated surface was washed with DDW 10 times, the resistance of all PPC-coated surfaces was almost unchanged (0.75, 1.38, 1.21, and 2.61 MΩ for PPC1, PPC2, PPC3, and PPC4, respectively), confirming the stability of the PPC coating on the surface. Moreover, contact angle analysis revealed that the hydrophilicity–hydrophobicity of the PPC-coated surfaces was the same, even after washing the surfaces 10 times (bare Si wafer: 52.3° to 52.3°, PPC1: 10.4° to 10.5°, PPC2: 11.3° to 11.3°, PPC3: 12.5° to 12.6°, and PPC4: 40.5° to 40.5°), indicating that the PPC stably covered the surface even after being exposed to frequent washing (Figure 3b).

### 3.3. Performance of the PPC Shampoo at Hair Protection and Restoration

The synthesized PPC shampoo samples were then applied to hair samples to evaluate their ability to provide hair protection and improve the mechanical properties of the hair. Static hair is a common problem that arises due to the triboelectric effect, particularly in dry and cold conditions. Hair with a high electrostatic charge causes inconvenience because the hair repels or entangles each other, often leading to brittleness, a dull color, and damage. Thus, the antistatic function of shampoos is crucial for reducing these effects by dissipating excess electrostatic charges on the hair [27]. The antistatic effect of the PPC shampoos was evaluated by applying each shampoo containing different PPC samples (1–4) to dry hair samples and placing the treated hair close to a rubbed balloon. A shampoo without additional PPC was used as a control. As shown in Figure 4, hair treated with the non-PPC shampoo displayed an evident static interaction with the balloon. In contrast, applying PPC shampoo effectively reduced the static effect on hair. Among the PPC shampoos, PPC1 demonstrated the most substantial antistatic effect, because of the low resistivity compared to the other PPCs. As indicated by previous source meter measurements, this enabled effective electrostatic charge dissipation on the PPC1 shampoo-treated hair. Based on these experimental data, we chose PPC1-supplemented shampoo as the optimum formula and further evaluated its hair-protection and -repair functions. Furthermore, the UV protection capability of PPC1-supplemented shampoo was confirmed by measuring the color change in dye-coated hair samples treated with shampoo alone (without PPC1, control) and with PPC1-supplemented shampoo after exposing the treated hair sample to UV irradiation for seven days. As shown in Appendix A, a substantial change in the color value (a*) was observed in the control hair model (day 0 = 9.8, day 7 = 7.2), while the hair sample treated with PPC1-supplemented shampoo showed a negligible change in color, even after 7 days of UV exposure (day 0 = 9.6, day 7 = 9.5). These results indicated the role of PPC1 in the shampoo in protecting hair from damage due to UV irradiation. PPC1 was capable of absorbing a broad spectrum of UV radiation owing to its graphene-like π-π stacking structure, which preserved the hair color from deterioration caused by continuous UV exposure.

Prolonged UV exposure and heat (from the sun), hair dyeing and bleaching treatments (containing oxidants such as H_2_O_2_), and regular hair dryer usage are the major factors that cause hair damage and the deterioration of hair strength. To address this issue, the hair recovery capabilities of the PPC1 shampoo were evaluated by comparing pre- and post-treatment damaged hair.

For this study, damaged hair samples were generated by exposing the hair to various concentrations of H_2_O_2_ (0, 6, and 9%). The damaged hair was treated with PPC1 shampoo (samples 1–3) and shampoo without PPC1 (control). The level of hair repair was observed by SEM imaging (Figure 5a). The damaged hair samples treated with the control shampoo showed no difference or further repair of the hair cuticle between pre- and post-treatment. For PPC1 shampoo-treated damaged hair samples (samples 1–3), the degree of deterioration of the hair cuticle before applying the PPC1 shampoo increased with increasing concentrations of H_2_O_2_ (0–9%), illustrating the damaging effect of H_2_O_2_ on hair. After using the PPC1 shampoo, the damaged hair cuticle was restored, and the hair was smoother than the hair treated with the control shampoo, indicating the capability of the PPC1 shampoo to recover hair from damage. These effects were due to the presence of a TA/GA/CA complex with abundant hydrogen bonding and π-π stacking, effectively promoting hair restoration and improvement. Confocal imaging revealed a difference between the hair samples treated with the control shampoo (without PPC1) and those treated with the PPC1 shampoo. As shown in Figure 5b, the control shampoo showed no fluorescence even after its application. In contrast, the hair treated with the PPC1 shampoo exhibited blue fluorescence, indicating the presence of the TA/GA/CA complex on the entire hair cuticle; thus, demonstrating total hair protection by the PPC1 shampoo.

Furthermore, the effect of PPC1 shampoo treatment on the hair mechanical properties of damaged hair samples (samples 1–3) was determined using a universal testing machine (UTM) as shown in Figure 6. The tensile stress and strain of hair sample 1 (H_2_O_2_, 0%) were significantly increased after treatment with PPC1 shampoo (6.03 N/mm^2^ and 45.47%, respectively), compared to the pre-treatment values (4.21 N/mm^2^ and 32.84%, respectively). Even in the presence of higher H_2_O_2_ concentrations in sample 2 (6%) and sample 3 (9%), the PPC1 shampoo was able to enhance the tensile stress and strain of the damaged hair model in (sample 2: 3.46 N/mm^2^ to 4.67 N/mm^2^ and 20.97% to 40.28%, respectively; sample 3: 3.05 N/mm^2^ to 3.79 N/mm^2^ and 13.63% to 28.08%, respectively). Interestingly, pre-treated sample 3 showed more extensive damage than samples 1 and 2, as reflected by the lower tensile strain because of the high concentration of H_2_O_2_. However, sample 3 demonstrated the greatest enhancement of mechanical properties at post-treatment (2.06 times the tensile strain of the pre-treatment sample) compared to samples 1 (1.38 times the tensile strain of the pre-treatment sample) and 2 (1.92 times the tensile strain of the pre-treatment sample). In contrast to hair treated with the PPC1 shampoo, control shampoo-treated hair (without PPC1) did not show any improvement in tensile strength, confirming the capability of the PPC1 shampoo to restore the mechanical strength of damaged hair owing to the abundance of hydrogen bonding and π-π stacking in the TA/GA/CA complex. Additionally, an improvement in mechanical strength was observed in the hair samples damaged by heat exposure. As shown in Appendix A, a considerable difference in tensile strength was observed between heat-damaged hair before (2.08 N/mm^2^, 14.45%) and after applying the PPC1 shampoo (4.71 N/mm^2^, 54.45%). These data confirm the capability of the PPC1 shampoo to enhance the mechanical properties of damaged hair.

Additionally, the ROS-scavenging efficiency of the PPC1 shampoo was measured using a DPPH assay (Figure 7). The shampoo without PPC showed no scavenging ability, as indicated by almost 0% DPPH inhibition, whereas the PPC1 shampoo demonstrated excellent ROS scavenging activity, with approximately 58% DPPH inhibition. This DPPH-inhibition efficiency was found to be close to that of PPC1 alone (69%), confirming the role of PPC1 in the shampoo in scavenging ROS. This was also supported by confocal imaging data of in vitro ROS staining (Appendix A), which showed that PPC1 completely scavenged ROS in HeLa cells after 12 h. Based on these data, supplementation of shampoos with PPC1 (TA/GA/CA) provides excellent antistatic effects, the enhancement of mechanical properties, and ROS-scavenging activity, which are crucial for hair protection and restoration.

## 4. Conclusions

We successfully synthesized a PPC-supplemented shampoo to protect hair from UV radiation, dyeing, bleaching, and drying. Simultaneously, we restored damaged hair by enhancing its mechanical properties. The combination of TA, GA, and CA provided integrated advantages, such as electroconductivity and durability, by increasing the mechanical strength, UV protection, and ROS-scavenging properties. TA, GA, and CA (PPC1) complexation produced a graphene-like structure via hydrogen bonding and π-π stacking interactions, which are essential for hair protection and repair. PPC1 demonstrated UV protection owing to its ability to absorb light across a broad spectrum, particularly in the UV region (200–380 nm). Additionally, it showed excellent coating stability, even after frequent washing (up to 10 times) owing to the strong adhesiveness of the polyphenol compounds. The application of a shampoo containing PPC1 to hair samples promoted an antistatic effect on the hair samples, which was triggered by an increase in the conductivity of the hair cuticle in the presence of PPC1 and effectively dissipated the electrostatic charges on the hair. Moreover, the PPC1 shampoo recovered, strengthened, and smoothened hair cuticles damaged by H_2_O_2_, with up to a 2.06-fold increase in tensile strain between post-treatment and pre-treatment hair. Additionally, the PPC1 shampoo decreased the levels of ROS owing to its high ROS-scavenging efficiency (approximately 60% DPPH inhibition). Hence, synthesizing/supplying a shampoo with PPC offers a new method for simultaneous hair protection and repair. Additionally, it has the potential for use in future hair restoration technologies.

## Data Availability

Data available on request from the authors.

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
