# Peer review of "Protection and Restoration of Damaged Hair via a Polyphenol Complex by Promoting Mechanical Strength, Antistatic, and Ultraviolet Protection Properties"

_biomimetics, 2023, doi:10.3390/biomimetics8030296_

Round 1

Reviewer 1 Report

General comment: The manuscript developed a spider silk-inspired hair-coating polyphenol complex (PPC) that shows UV-blocking properties, antistatic features, and the capability to enhance the mechanical strength of damaged hair. It is well written and suitable experimental approaches were used to response their questions. The manuscript is of general importance to a wide audience. However, I think there are some points that should be clarified before accepted this work for its publication.

Comment 1: It's lack of persuasiveness that proving the relative content of hydrogen bonds only through FT-IR spectra.

Comment 2: In your experiment in Figure 3, it is better to add a blank control to prove that the synthesized substance has better effects.

Comment 3: In Figure 5b, what is the treatment for hair to emit blue fluorescence, please provide a detailed description.

Comment 4: It’s beat to present clearly the nanostructure of the substance you synthesized.

Comment 5: More experimental data is needed to conclude that your substance can help hair resist ultraviolet radiation.

English language and style used throughout the manuscript needs to be thoroughly revised and proofed by a native speaker, currently there are some grammatical errors and some sentences are hard to understand. 

Author Response

Reviewer 1

General comment: The manuscript developed a spider silk-inspired hair-coating polyphenol complex (PPC) that shows UV-blocking properties, antistatic features, and the capability to enhance the mechanical strength of damaged hair. It is well written and suitable experimental approaches were used to response their questions. The manuscript is of general importance to a wide audience. However, I think there are some points that should be clarified before accepted this work for its publication.

1. Comment 1: It's lack of persuasiveness that proving the relative content of hydrogen bonds only through FT-IR spectra.

Thank you for the valuable comment. Other than FT-IR result, we added explanation regarding the hydrogen bonding based on UV-vis spectra (Figure 1a) and the newly supplied TGA (Figure S2) on page 5 lines 193-194 and page 6 lines 219-223, respectively. The blue shift in UV-visible spectra was observed and the degradation temperature was increased to a higher temperature, which influence by the effect of intermolecular interaction after forming the complex. We added the new references for TGA measurement (ref no. 34, 35, and 36).

For UV-vis on page 5 lines 193-194, “The shifting of the π-π* absorption peak also indicated the effect of intermolecular interactions, such as hydrogen bonding and π-π stacking.”

For TGA data on page 6 lines 219-223, “This was further supported by TGA data, which showed an increase in the degradation temperature of CA (217–240 °C) [34], GA (325–400 °C) [35], and TA (380–600 °C) alone [36], compared to when they were part of the synthesized PPC1 complex (203–289 °C, 292–493 °C, and 400–646 °C for CA, GA, and TA, respectively). This was related to the hydro-gen bonding in PPC1 (Figure S2).”

2. Comment 2: In your experiment in Figure 3, it is better to add a blank control to prove that the synthesized substance has better effects.

  • Thank you for the advice, we added the resistance (0.874 MΩ) and the contact angle (52.3o) of the blank Si wafer in revised Figure 3 and mention it in the revised version of the manuscript on page 8 lines 235-238 and lines 244-248.

For page 8 lines 235-238, “Based on source meter measurements (Figure 3a), the unwashed PPC1-coated surface showed lower resistance (0.72 MΩ) than the unwashed surfaces coated with PPC2 (1.33 MΩ), PPC3 (1.11 MΩ), or PPC4 (2.41 MΩ) and the bare Si wafer (0.874 MΩ).”

For page 8 lines 244-248, “Moreover, contact angle analysis revealed that the hydrophilicity-hydrophobicity of the PPC-coated surfaces was the same, even after washing the surfaces 10 times (bare Si wafer: 52.3° to 52.3°, PPC1: 10.4° to 10.5°, PPC2: 11.3° to 11.3°, PPC3: 12.5° to 12.6°, and PPC4: 40.5° to 40.5°), indicating that the PPC stably covered the surface even after being exposed to frequent washing (Figure 3b).”

3. Comment 3: In Figure 5b, what is the treatment for hair to emit blue fluorescence, please provide a detailed description.

Thank you for your valuable comment, we provided the photoluminescence (PL) spectra of the TA/GA/CA (PPC) in the supporting information (Figure S1) and added explanation on page 5 lines 200-202. Based on the PL spectra, it could be expected that the TA/CA/GA exhibited blue fluorescent.

“Additionally, the photoluminescence spectra showed that PPC1 emitted blue fluorescence due to the presence of the π-π stacking structure, which served as a fluorophore (Figure S1).”

4. Comment 4: It’s beat to present clearly the nanostructure of the substance you synthesized.

  • As suggestion reviewer comment, we revised the Scheme 1a of the nanoparticles structure in the revised manuscript.

Scheme 1. a) Schematic illustration of polyphenol complex (PPC) synthesis using tannic acid (TA), gallic acid (GA), and caffeic acid (CA).

5. Comment 5: More experimental data is needed to conclude that your substance can help hair resist ultraviolet radiation.

Thank you for the advice, we have performed additional experiment to investigate the performance of the PPC shampoo against ultraviolet (UV) as shown in Figure S3 and explain on page 9 lines 272-283. The experiment was conducted by irradiating the red-dyed hair sample without and with treatment with PPC1 shampoo and the degradation of color was investigated with colorimeter (Konica, Minolta, Japan).

“Furthermore, UV protection capability of PPC1-supplemented shampoo was confirmed by measuring the color change of dye-coated hair samples treated with shampoo alone (without PPC1, control) and with PPC1-supplemented shampoo after exposing the treated hair sample to UV irradiation for seven days. As shown in Figure S3, a substantial change in the color value (a*) was observed in the control hair model (day 0 = 9.8, day 7 = 7.2), while the hair sample treated with PPC1-supplemented shampoo showed a negligible change in color, even after 7 days of UV exposure (day 0 = 9.6, day 7 = 9.5). These results indicated the role of PPC1 in the shampoo in protecting hair from damage due to UV irradiation. PPC1 was capable of absorbing a broad spectrum of UV radiation owing to its graphene-like π-π stacking structure, which preserved the hair color from deterioration caused by continuous UV exposure.

The detail experimental method for UV protection performance was added on page 4 lines 136-142.

“UV protection analysis was performed by irradiating the hair sample at 302 nm for 72 h. Initially, the hair samples were dyed using a red dye (Allura Red AC 40) as a marker of color degradation. The dye was initially applied to the hair 30 times (all sides of the hair) and allowed to coat the hair for 30 min. The hair was then washed with running water and left to naturally dry for 1 h. Thereafter, the PPC shampoo was applied to the hair, and the hair was irradiated UV light and investigated using a colorimeter (Konica Minolta).”

6. English language and style used throughout the manuscript needs to be thoroughly revised and proofed by a native speaker, currently there are some grammatical errors, and some sentences are hard to understand. 

We change the English using native speaker help, and the certificate of English editing has been added to the cover letter.

Reviewer 2 Report

Dear Authors,

After careful reading, below, are the comments/suggestions in regard to your manuscript.

- the term "blocking" is not acceptable for a property of sunscreen effect or photoprotection. It must be substituted by an appropriate terminology.

- please, considering the hair shaft a dead material, how could it develop a self-recovery activity?

- Table 1 was misplaced, wasn't it?

- please, describe the qualitative composition, at least, of the shampoo sample.

- how were the hair tresses or hair shafts treated before investigation? 

- please, present clearly the results from the nanostructure free from the actives. 

- how was the UV protection determined with the hair samples?

Overall, as described, this manuscript presented a level of difficult to achieve high comprehension from reading. Several methods were poorly described. The nano without the acids was difficult to find along the text.

Author Response

Reviewer 2

After careful reading, below, are the comments/suggestions in regard to your manuscript.

1. the term "blocking" is not acceptable for a property of sunscreen effect or photoprotection. It must be substituted by an appropriate terminology.

Thank you for the comment, we replace the term “blocking” into “protection” to avoid misunderstanding, such as in the title part.

“Protection and Restoration of Damaged Hair via a Polyphenol Complex by Promoting Mechanical Strength, Antistatic, and Ultraviolet Protection Properties”

2. please, considering the hair shaft a dead material, how could it develop a self-recovery activity?

Thank you for the question. The mechanical strength of hair shaft is contributed by the presence of keratin, a main component of hair cortex. The chemical hair treatment such as hair dyeing and bleaching, also prolong exposure to UV radiation and heat from hair drying process can cleave keratin on the hair shaft, which deteriorates the hair’s mechanical strength (Ref: 10.1016/j.btre.2018.e00288, 10.1111/j.1365-2133.2012.10862.x). To enhance hair recovery and prevent hair from excess damage, it is important to protect the hair by applying the improvement materials, in our case is PPC complex, on the hair for not only improving the tensile mechanics but also protecting hair from external stimuli. Owing to the presence of PPC, which possess rich hydrogen bond, UV protection effect and reactive oxygen species (ROS) scavenging ability, the PPC1 shampoo treated hair can restore the damaged hair and amplify the hair's mechanical strength and morphology.

3. Table 1 was misplaced, wasn't it?

As per the reviewer suggestion, we adjusted the position of the Table 1.

4. please, describe the qualitative composition, at least, of the shampoo sample.

Regarding the shampoo formulation, we used the MODAMODA Pro-Change Black ShampooÒ that contained black cherry, black sesame, black berry, black cumin, and black truffle (page 3 line 86-88). Moreover, we can provide the ratio of between the shampoo and the additive materials (PPC complex). We have stated in the manuscript that the ratio between the shampoo and the PPC complex is estimated at 250:1 (30 g: 120.5 mg) as mentioned in Material and Methods section on page 3 lines 110-111.

For page 3 line 86-88, “The MODAMODA Corporation (Seoul, South Korea) supplied the Pro-Change Black ShampooÒ (base shampoo containing black cherry, black sesame, blackberry, black cumin, and black truffle).”

For page 3 lines 110-111, “Thereafter, the PPC was added to 30 g of shampoo, mixed for 30 min at 40 ℃, and labeled as PPC shampoo.”

5. how were the hair tresses or hair shafts treated before investigation? 

Thank you for the suggestion, we have added the detail of the methods in the revised manuscript on page 4 lines 131-142.

“For the hydrogen peroxide (H2O2) condition, the hair samples were immersed in H2O2 solutions of different concentrations (0, 6, and 9%) for 12 h and then washed with running water for 1 min before PPC shampoo treatment. For the heat treatment, the hair samples were dried 30 times (one-directional swipe) with a hair straightener at a temperature of 220 ℃ (44 W) before PPC shampoo treatment. UV protection analysis was performed by irradiating the hair sample at 302 nm for 72 h. Initially, the hair samples were dyed using a red dye (Allura Red AC 40) as a marker of color degradation. The dye was initially ap-plied to the hair 30 times (all sides of the hair) and allowed to coat the hair for 30 min. The hair was then washed with running water and left to naturally dry for 1 h. Thereafter, the PPC shampoo was applied to the hair, and the hair was irradiated UV light and investigated using a colorimeter (Konica Minolta).”

6. please, present clearly the results from the nanostructure free from the actives.

Thank you for your comment. In our experiments, we have shown the results of shampoo without additional PPC1 (actives ingredient) as a comparison for PPC1-supplemented shampoo, such as in Figure 4 (page 8), Figure 5 (page 9), and Figure 6 (page 10). From those data, we can conclude that the presence of active PPC1 provides remarkable antistatic effect, hair protection, and enhancement of hair’s mechanical strength compared to the shampoo only (without PPC1).

7. how was the UV protection determined with the hair samples?

Thank you for the advice, we have performed additional experiment to investigate the performance of the PPC shampoo against ultraviolet (UV) as shown in Figure S3 and explain on page 9 lines 272-283. The experiment was conducted by irradiating the red-dyed hair sample without and with treatment with PPC1 shampoo and the degradation of color was investigated with colorimeter (Konica, Minolta, Japan).

“Furthermore, UV protection capability of PPC1-supplemented shampoo was confirmed by measuring the color change of dye-coated hair samples treated with shampoo alone (without PPC1, control) and with PPC1-supplemented shampoo after exposing the treated hair sample to UV irradiation for seven days. As shown in Figure S3, a substantial change in the color value (a*) was observed in the control hair model (day 0 = 9.8, day 7 = 7.2), while the hair sample treated with PPC1-supplemented shampoo showed a negligible change in color, even after 7 days of UV exposure (day 0 = 9.6, day 7 = 9.5). These results indicated the role of PPC1 in the shampoo in protecting hair from damage due to UV irradiation. PPC1 was capable of absorbing a broad spectrum of UV radiation owing to its graphene-like π-π stacking structure, which preserved the hair color from deterioration caused by continuous UV exposure”

Reviewer 3 Report

In this manuscriptJeong Won et al, investigated the impact of a combination of these three polyphenols (PPC) in a shampoo formulation to protect hair against UV radiation. The overall quality of the paper is good, but with important space for improvement. The paper's methodology is clear, focusing on the mechanical, chemical, and biological properties of three types of polyphenols: TA, GA, and CA.

In the introduction (line 38), the authors mentioned that "exposure to UV light induces photodegradation of hair proteins." It is now evident that this process occurs through the accumulation of ROS-induced modifications, mainly  protein oxidation, such as carbonylation. To support this statement, the following reference should be cited in the article: "Protein Carbonylation as a Reliable Read-Out of Urban Pollution Damage/Protection of Hair Fibers" by Cavagnino et al., 2022 (DOI: 10.3390/cosmetics9050098).

In the "Materials and Methods" section, a clear procedure on  how the formulation containing PCC was applied should be added to enhance the understanding of the methodology and to identify the stressors to which the treated hair was exposed. For example, in Figure 5, it is mentioned that the hair samples were exposed to H2O2, but this step was not reported in the materials and methods.

On line 252, it is indicated that hair samples were exposed to H2O2 at concentrations of 0.6% and 9% to mimic hair damage caused by various stressors. It is unfortunate that the protective effect of embedded PCC in the shampoo was not tested against different types of stressors, such as UV or heat (hair dryer). Indeed, the UV protective effect could be linked to the first part of this study, which investigated the UV blocking effect of free PCC complex using UV-visible spectroscopy.

To visualize and quantify the recovery of hair fibers structural integrity following hair damage, the authors should consider performing additional experiments to support this concept (cuticle structural integrity

Biochemical analysis are only performed in tubo. Ex vivo analyses of anti-oxidant effects on hair fibers are desirable to support the conclusiosn. In addition,  the DPPH assay might not be the most suitable method for quantifying the effects of products under stress. Currently, the state-of-the-art research focuses on protein modifications induced by oxidative stressors such as H2O2, UV radiation, and urban pollution. These stressors are known to mediate the carbonylation of proteins, which serves as a reliable biomarker for assessing the early events of hair damage/protection. Statistical analyses are missing and must be performed to validate the data.

Revision is needed

Author Response

Reviewer 3

In this manuscript Jeong Won et al, investigated the impact of a combination of these three polyphenols (PPC) in a shampoo formulation to protect hair against UV radiation. The overall quality of the paper is good, but with important space for improvement. The paper's methodology is clear, focusing on the mechanical, chemical, and biological properties of three types of polyphenols: TA, GA, and CA.

1. In the introduction (line 38), the authors mentioned that "exposure to UV light induces photodegradation of hair proteins." It is now evident that this process occurs through the accumulation of ROS-induced modifications, mainly protein oxidation, such as carbonylation. To support this statement, the following reference should be cited in the article: "Protein Carbonylation as a Reliable Read-Out of Urban Pollution Damage/Protection of Hair Fibers" by Cavagnino et al., 2022 (DOI: 10.3390/cosmetics9050098).

Thank you for the comment, we added the reference to support our finding (new reference 6) on page 1 line 36-38.

2. In the "Materials and Methods" section, a clear procedure on how the formulation containing PCC was applied should be added to enhance the understanding of the methodology and to identify the stressors to which the treated hair was exposed. For example, in Figure 5, it is mentioned that the hair samples were exposed to H2O2, but this step was not reported in the materials and methods.

Thank you for the suggestion, we have added the detail experiment for H2O2 treatment in the revised manuscript on page 4 line 131-134.

“For the hydrogen peroxide (H2O2) condition, the hair samples were immersed in H2O2 solutions of different concentrations (0, 6, and 9%) for 12 h and then washed with running water for 1 min before PPC shampoo treatment.”

3. On line 252, it is indicated that hair samples were exposed to H2O2 at concentrations of 0.6% and 9% to mimic hair damage caused by various stressors. It is unfortunate that the protective effect of embedded PCC in the shampoo was not tested against different types of stressors, such as UV or heat (hair dryer). Indeed, the UV protective effect could be linked to the first part of this study, which investigated the UV blocking effect of free PCC complex using UV-visible spectroscopy.

Thank you for the valuable comment, we perform the PPC shampoo effect to the UV-treated hair using color change (colorimeter) and hair-dried hair universal testing machine (UTM) as shown in Figure S3 and Figure S4, respectively. The UV protection experiment was conducted by irradiating the red-dyed hair sample without and with treatment with PPC1 shampoo and the degradation of color was investigated with colorimeter (Konica, Minolta, Japan). The result of the colorimetric analysis (Figure S3) was mentioned on page 9 lines 272-283 of the revised manuscript.

“Furthermore, UV protection capability of PPC1-supplemented shampoo was confirmed by measuring the color change of dye-coated hair samples treated with shampoo alone (without PPC1, control) and with PPC1-supplemented shampoo after exposing the treated hair sample to UV irradiation for seven days. As shown in Figure S3, a substantial change in the color value (a*) was observed in the control hair model (day 0 = 9.8, day 7 = 7.2), while the hair sample treated with PPC1-supplemented shampoo showed a negligible change in color, even after 7 days of UV exposure (day 0 = 9.6, day 7 = 9.5). These results indicated the role of PPC1 in the shampoo in protecting hair from damage due to UV irradiation. PPC1 was capable of absorbing a broad spectrum of UV radiation owing to its graphene-like π-π stacking structure, which preserved the hair color from deterioration caused by continuous UV exposure. “

The post-treatment hair showed an enhance in the mechanical performance (Figure S4) in the tensile stress and strain from 08 N/mm2, 14.45% to 4.71 N/mm2, 54.45% as explain on page 11 lines 335-340.

“Additionally, an improvement in mechanical strength was observed in the hair samples damaged by heat exposure. As shown in Figure S4, a considerable difference in tensile strength was observed between heat-damaged hair before (2.08 N/mm2, 14.45%) and after applying the PPC1 shampoo (4.71 N/mm2, 54.45%). These data confirm the capability of the PPC1 shampoo to enhance the mechanical properties of damaged hair.”

4. To visualize and quantify the recovery of hair fibers structural integrity following hair damage, the authors should consider performing additional experiments to support this concept (cuticle structural integrity)

We have already provided the visualization of hair recovery from SEM imaging at Figure 5a at page 9 with explanation at page 10 lines 295-307. For a quantitative analysis regarding hair recovery, we have conducted UTM measurement as shown in Figure 6 at page 11 with explanation at page 10-11 lines 317-335. We have also added the data of heat-damaged hair recovery after applying PPC1-supplemented shampoo in Figure S4 Supporting Information at page S-6, with explanation at page 11 lines 335-340.

“Additionally, an improvement in mechanical strength was observed in the hair samples damaged by heat exposure. As shown in Figure S4, a considerable difference in tensile strength was observed between heat-damaged hair before (2.08 N/mm2, 14.45%) and after applying the PPC1 shampoo (4.71 N/mm2, 54.45%). These data confirm the capability of the PPC1 shampoo to enhance the mechanical properties of damaged hair.”

5. Biochemical analyses are only performed in tubo. Ex vivo analyses of antioxidant effects on hair fibers are desirable to support the conclusion. In addition, the DPPH assay might not be the most suitable method for quantifying the effects of products under stress. Currently, the state-of-the-art research focuses on protein modifications induced by oxidative stressors such as H2O2, UV radiation, and urban pollution. These stressors are known to mediate the carbonylation of proteins, which serves as a reliable biomarker for assessing the early events of hair damage/protection. Statistical analyses are missing and must be performed to validate the data.

Thank you for your comment, we performed in vitro investigation of the cellular ROS using confocal imaging. The HeLa cells was treated using PPC1 and then dyed with H2DCFDA dyes and observed with confocal microscope as shown in Figure S5. The explanation was added on page 11 lines 350-352. The PPC1-treated cells showed high scavenging performance as displayed by no green fluorescent on the cells.

“This was also supported by confocal imaging data of in vitro ROS staining (Figure S5), which showed that PPC1 completely scavenged ROS in HeLa cells after 12 h.”

The experimental detail was added in the supporting information on page S2.

“1. Experimental Section

  • In vitro investigation of ROS scavenging capabilities

For optical imaging, HeLa cells with a concentration of 105 cells/well were cultured on a 24-well plate and placed in a humidified incubator (5% CO2 atmosphere) for 12 h at 37°C. Then, the solution media was changed with media containing TA/GA/CA at a concentration of 1.6 mg/mL and incubated at designated times (0, 3, 6, and 12 h). The cells were detached using trypsin-EDTA, centrifuged, and washed with PBS pH 7.4 before being stained with 2′,7′-dichlorodihydrofluorescein diacetate (H2DCFDA) and imaged with a confocal microscope (magnification, 40×).”

For statistical analysis, we added the number of experiment (n value) and input the statistical significance (p value) of the DPPH measurement.

Round 2

Reviewer 2 Report

Dear Authors,

Thank you for responding all questions from peer-review. Please, just add the qualitative composition of the shampoo, only the actives were described.

Author Response

Reviewer 2

Thank you for responding all questions from peer-review. Please, just add the qualitative composition of the shampoo, only the actives were described.

  • Thank you for your comment. We have added the qualitative composition of the shampoo as described at page 3 line 86-94.

“The MODAMODA Corporation (Seoul, South Korea) supplied the Pro-Change Black ShampooÒ (containing water, sodium C14-16 olefin sulfonate, disodium laureth sulfosuccinate, glycerin, sodium cocoyl isethionate, lauryl hydroxysultaine, butylene glycol, lauryl glucoside, 1-2-hexanediol, Sesamum indicum (sesame) seed extract, Morus nigra fruit extract, Tuber melanosporum extract, Nigella sativa seed extract, Prunus serotina (wild cherry) fruit extract, Morus alba bark extract, niacinamide, Camelia sinensis leaf extract, 1,2,4-trihydroxybenzene, salicylic acid, citric acid, Mentha piperita (peppermint) oil, sodium chloride, PPG-3 caprylyl ether, guar hydroxypropyltrimonium chloride, panthenol, polyquartenium-10, ethylhexylglycerin, caprylyl glycol, menthol, caramel).”

Reviewer 3 Report

The authors addressed most of my concerns

Minor revision is needed

Author Response

Reviewer 3

Comments on the Quality of English Language: Minor revision is needed.

Thank you for your comment. The English writing including the grammatical structure in this manuscript have been carefully checked by professional English editing service (native speaker). We have attached the certificate of English editing in the Cover Letter.
